# Pomegranate Extract Augments Energy Expenditure Counteracting the Metabolic Stress Associated with High-Fat-Diet-Induced Obesity

**DOI:** 10.3390/ijms231810460

**Published:** 2022-09-09

**Authors:** Marina Reguero, Marta Gómez de Cedrón, Aranzazu Sierra-Ramírez, Pablo José Fernández-Marcos, Guillermo Reglero, José Carlos Quintela, Ana Ramírez de Molina

**Affiliations:** 1Molecular Oncology Group, IMDEA Food Institute, CEI UAM + CSIC, 28049 Madrid, Spain; 2NATAC BIOTECH, Electronica 7, 28923 Madrid, Spain; 3Metabolic Syndrome Group, IMDEA Food, CEI UAM + CSIC, 28049 Madrid, Spain; 4Production and Characterization of Novel Foods Department, Institute of Food Science Research CIAL, CEI UAM + CSIC, 28049 Madrid, Spain

**Keywords:** energy expenditure, thermogenesis, meta-inflammation, obesity, pomegranate extract

## Abstract

Obesity is associated to a low grade of chronic inflammation leading to metabolic stress, insulin resistance, metabolic syndrome, dislipidemia, cardiovascular disease, and even cancer. A Mediterranean diet has been shown to reduce systemic inflammatory factors, insulin resistance, and metabolic syndrome. In this scenario, precision nutrition may provide complementary approaches to target the metabolic alterations associated to “unhealthy obesity”. In a previous work, we described a pomegranate extract (PomE) rich in punicalagines to augment markers of browning and thermogenesis in human differentiated adipocytes and to augment the oxidative respiratory capacity in human differentiated myocytes. Herein, we have conducted a preclinical study of high-fat-diet (HFD)-induced obesity where PomE augments the systemic energy expenditure (EE) contributing to a reduction in the low grade of chronic inflammation and insulin resistance associated to obesity. At the molecular level, PomE promotes browning and thermogenesis in adipose tissue, reducing inflammatory markers and augmenting the reductive potential to control the oxidative stress associated to the HFD. PomE merits further investigation as a complementary approach to alleviate obesity, reducing the low grade of chronic inflammation and metabolic stress.

## 1. Introduction

Nowadays, the increase in life expectancy has favoured the appearance of numerous diseases related to ageing. Among them, metabolic disorders, such as obesity, are increasing exponentially, affecting a large part of the population worldwide. In addition, obesity aggravates the development and prognosis of other pathologies such as metabolic syndrome, diabetes, cardiovascular diseases, infections, and even cancer [1,2].

Considering the prevalence of obesity, numerous therapeutic alternatives to combat different aspects of obesity-associated metabolic alterations are currently being studied.

Recently, it has been proposed that augmenting energy expenditure (EE), by means of the induction of adaptive thermogenesis in the adipose tissues and the increase in the oxidative mitochondrial capacity of muscles, may contribute to alleviating the metabolic stress associated to unhealthy obesity. Bioactive compounds from natural sources or diet-derived ingredients have been demonstrated to ameliorate the metabolic and oxidative stress associated with obesity by targeting relevant pathways involved in the activation of mitochondrial uncoupling, the reduction of inflammation, the increase in mitochondrial oxidative capacity, and/or the increase in reductive power, among others. The interaction between metabolic organs, such as adipose tissue and muscles, has been shown to control the systemic metabolic and oxidative stress during obesity. Although there is a genetic susceptibility to the development of metabolic alterations, obesity may be preventable by means of diet-based interventions, including bioactive compounds and natural extracts. Many of the comorbidities associated with obesity are related to white adipose tissue (WAT) dysfunction, which promotes a low grade of chronic inflammation [3,4]. Additionally, although brown adipose tissue (BAT) is less susceptible to the development of local inflammation, alterations in its function also have important consequences on the systemic energy balance [5,6].

In this context, precision nutrition represents a very promising field of study to provide complementary treatments in the control of the metabolic alterations associated to chronic diseases. Molecular nutrition is based on the scientific knowledge of the functional effects of bioactive compounds present in the diet. Importantly, diet-based interventions should take into consideration not only the genetic and molecular alterations, but also the lifestyle and nutritional status of individuals. Plants constitute an unlimited source of bioactive compounds that can contribute to ameliorating the metabolic alterations and the prognosis of comorbidities associated to obesity [7]. With the understanding that the pathogenesis of many diseases involves multiple factors, the focus of drug discovery has shifted from the conventional “one target, one drug” model to a new “multi-target, multidrug” model. Thus, it has been shown that natural extracts, as a combination of bioactive compounds, may target simultaneously different pathways, which may improve the beneficial effects of the extract. Indeed, accumulating evidence from clinical studies supports the claim that natural extracts represent an efficient form of therapy in the control of complex diseases, such as obesity, metabolic syndrome, cardiovascular disease, type 2 diabetes, and cancer. In a previous work, we evaluated the functional effects of twenty plant-based extracts (from the Natac Group’s Library, https://natacgroup.com/products/extracts-portfolio/ (accessed on 27 July 2022)) on differentiated human adipocytes and myocytes [8]. Among them, PomE exerted strong effects on the activation of thermogenesis—lipolysis, β-oxidation, and uncoupling—and mitochondrial metabolism in in vitro cell models [8]. At the molecular level, PomE augmented the expression of *UCP1*, *UCP2*, *BMP8B*, *CKMT*, *AMPK*, and *PRDM16*, which are key regulators of mitochondrial function and thermogenesis in differentiated human adipocytes [9,10,11,12,13], as well the expression of key myokines such as *BDNF* in differentiated myocytes, which has been described as a paracrine effector of browning [14,15,16]. Importantly, PomE enhanced the mitochondrial respiration capacity and the uncoupling and proton leakage in adipocytes and myocytes in vitro.

Therefore, herein we aim to evaluate in vivo the potential of PomE to increase systemic energy expenditure by means of the activation of WAT browning and the function of muscles in a model of high-fat-diet (HFD)-induced obesity.

## 2. Results

In a previous work, we demonstrated that PomE augmented thermogenic markers and induced lipolysis, uncoupling and mitochondrial respiration in an in vitro model of differentiated human adipocytes (SGBS). In addition, PomE augmented the expression of key regulators of mitochondrial function and myokines implicated in the activation of browning of adipose tissue in human differentiated myocytes [8]. Therefore, herein we aimed to evaluate the in vivo effect of PomE to augment the systemic EE by means of the activation of browning and thermogenesis in a model of HFD-induced obesity.

### 2.1. Study Workflow

To evaluate the potential of PomE in alleviating the metabolic stress associated to obesity, a preclinical study of high-fat-diet (HFD)-induced obesity was designed. A total of 28 six-week-old male C57BL/6 mice of were divided into three different groups, namely group 1: ND-control (N = 6); group 2: HFD-control (N = 10); and group 3: HFD-PomE (N = 12). For the first 6 weeks prior to the intervention, mice from groups 2 and 3 were fed with HFD to be acclimated to this new condition and to cause a metabolic disturbance associated to the weight gain. Afterward, at the age of 12 weeks, the administration of the PomE or the vehicle started. PomE was dissolved in drinking water and administered by gastric gavage (0.1 g of the extract per Kg of mice), 3 days per week over the 12–14 weeks.

Figure 1 summarizes the study workflow and main biochemical and molecular determinations.

First, the weight and food intake of the animals after 6 weeks of ND or HFD prior the intervention study were monitored. As observed in Figure 2A, a significant increase in food intake in the ND-control group was observed compared to that of the HFD-control and HFD-PomE groups, demonstrating the satiating effect of the HFD. Then, the intervention study was initiated in the HFD group for 12–14 weeks. As expected, the HFD-control and HFD-PomE groups significantly increased their weight compared to that of the ND-control group (Figure 2B). Interestingly, the HFD-PomE group showed a significant increase in food intake compared to the HFD-control group. However, animals in HFD-PomE did not augment their weight compared to that of the HFD group. In addition, the weight of the epididymal white adipose tissues (eWAT) of animals in the HFD-PomE group did not increase compared to that of the HFD-control group (Appendix A), suggesting underlying mechanisms related to systemic metabolic adaptations to the HFD exposition.

### 2.2. PomE Extract Improves the Systemic Glucose Homeostasis and Insulin Sensitivity after HFD Induced Obesity

First, the prediabetic situation after 6 weeks of HFD compared to ND was confirmed. Thus, increased glucose plasma levels and HOMA/IR index were found in the HFD group compared to the ND group (Appendix A). Then, the intervention with PomE (HFD-PomE) or vehicle (HFD-control) in combination with HFD was initiated. In addition, a normal diet group (ND group) was maintained for comparison. After twelve weeks of intervention, no significant differences were found in the maximum peaks of glycemia evaluated in the glucose tolerance test (GTT) nor in the insulin tolerance test (ITT) curves between HFD-PomE and HFD groups, although a slight reduction was found in the area under the curve (AUC) in the HFD-PomE group compared to the HFD group. Interestingly, fasting insulin was significantly lower in the HFD-PomE group compared to that of the HFD group. Importantly, the calculated HOMA/IR index in the HFD-PomE group was significantly reduced compared to the HFD group (*p*-value 0.047), although without recovering the levels of the ND group (Figure 3).

### 2.3. PomE Augments the Systemic Energy Expenditure after HFD-Induced Obesity

To elucidate the underlying mechanisms implicated on the improved insulin sensitivity in the HFD-PomE group compared to the HFD-control group, and taking into consideration that PomE augmented thermogenic and browning markers in an in vitro model of differentiated human adipocytes, we next applied indirect calorimetry to determine the volume of oxygen (VO_2_), volume of CO_2_ (VCO_2_), and energy expenditure parameters. As shown in Figure 4, a significant increase in VO_2_ (*p*-value 0.003) and VCO_2_ (*p*-value 0.01), as well as in EE (*p*-value 0.002), were found in the HFD-PomE group compared to that of the HFD-control group. No differences were observed in food intake and drinking water, nor in the activity of the animals in the HFD-PomE and HFD-control groups (Appendix A). The highest levels were detected in the dark phase, corresponding to the nocturnal phase of activity in mice, and the lowest values during midday (not shown).

### 2.4. PomE Preserves the Body Temperature after Cold Exposition

As previous results indicated that PomE augments the systemic energy expenditure, and to determine if activation of thermogenesis may be implicated, we next evaluated the role of PomE in the control of body temperature after cold exposition. For this, animals were exposed to 4 °C temperature for 24 h after a previous period of 72 h of acclimatation at 18 °C. As shown in Figure 5, PomE treatment preserved the body temperature compared to that of the HFD group. It is important to highlight that in both groups, the body temperature was drastically reduced from the first hours of exposure to cold, being very difficult to revert this thermal change in the animals just only with PomE treatment in such a short period of time (24 h of extreme cold). However, the temperature was improved significantly in the HFD-PomE group compared to that of the HFD-control group (*p*-value 0.0417). Therefore, this result suggests that PomE extract has a role in the regulation of body temperature after cold exposition by means of the activation of thermogenesis.

### 2.5. PomE Augments the Expression of Genes Implicated in Browning of WAT In Vivo

To clarify underlaying mechanisms implicated in PomE extract protection from metabolic stress after a HFD exposition, the different types of fat—epidydimal white adipose tissue (eWAT), inguinal adipose tissue (iWAT), brown adipose tissue (BAT)—and muscle tissues were saved for the analysis of the expression of genes related to thermogenesis, mitochondrial uncoupling, mitochondrial oxidative capacity, mitochondrial biogenesis, oxidative stress, and β-oxidation, among others.

Although we did not find differences in the expression levels of *UCP1* in the eWAT tissues of HFD-PomE and HFD-control groups (Figure 6), the expression of *UCP1* was augmented in the iWAT—which is the WAT susceptible to progress into beige—of the HFD-PomE group compared to that of the HFD-control group under cold exposition (Figure 6). As thermogenesis and its regulation not only depend on mitochondrial uncoupling proteins [17,18], we also analysed the expression of genes related to mitochondrial biogenesis and oxidative capacity. PomE induced the expression of key promoters of AT browning such as *BMP8B* in iWAT, both at normal temperature and under cold exposition [9,19,20,21]. In addition, *AMPK* and *SIRT1* genes were significantly increased in the iWAT of the HFD-PomE group compared to the HFD-control group under cold stress (Figure 6). AMPK activation is associated to SIRT1 activation controlling many metabolic functions, including differentiation of cultured adipocytes, and it has been extensively associated to the promotion of browning in HFD-induced obesity [22,23,24].

### 2.6. PomE Modulates the Expression of Inflammatory Myokines in Muscle Tissues

As obesity promotes a low grade of chronic inflammation, and due to the observed effects of PomE in the improved insulin sensitivity and the increased on EE and thermal temperature after cold exposition, we next evaluated the effects of PomE on markers of inflammation after HFD exposition.

Although no significant changes were found in the expression of *IL6* in the HFD-PomE group compared to the HFD-control group, an increase was observed in the expression of *IL17RA* in the iWAT of the animals when exposed to cold. This result suggests a role of PomE in a better control of body temperature after cold stress by means of the increase in *IL17RA* intracellular signalling, which has been extensively described to activate AT thermogenesis [25].

Inflammatory myokines such as IL6 produced by muscles have been demonstrated to promote lipolysis and browning of AT in a paracrine manner [26,27]. As shown in Figure 7, the expression levels of *IL6* in the muscles of the HFD-PomE group after cold exposition augmented significantly compared to that of the HFD-control group, suggesting a crosstalk between muscles and AT in the promotion of lipolysis.

### 2.7. PomE Affects the Expression of Molecular Targets Involved in Glucose and Redox Homeostasis

As PomE improved insulin sensitivity, we next analysed the expression levels of genes related to glycolysis and redox homeostasis in eWAT and iWAT tissues and in skeletal muscles, both under normal conditions and after cold exposition.

As shown in Figure 8, a significant positive modulation was found in the gene expression of glucose-6-phosphate dehydrogenase (*G6PDH*) in white muscles of HFD-PomE group compared to that of the HFD-control group after cold exposition. A similar tendency, although not significant, was observed in the absence of cold stress. Moreover, a positive tendency was observed in the expression levels of phosphofructokinase (*PFK*) in the BATs of the HFD-PomE group compared to the HFD group, both under normal and cold temperature.

These results suggest that PomE may reduce the negative impact of muscle oxidative stress, increasing the reductive power by means of increased expression of G6PDH, a key target in the formation of the reduced nicotinamide adenine dinucleotide (NADH) [28,29]. NADH’s main function is the exchange of electrons and protons and the production of energy in all cells, which is necessary for proper muscle recovery after physical exercise [30,31]. PomE may contribute to maintaining the redox balance, as has been previously reported with other extracts of this fruit [32,33,34,35,36].

In Figure 9 it is shown main biological activities of PomE to diminish the metabolic stress associated to HFD induced obesity. 

## 3. Discussion

Obesity is associated with a low grade of chronic inflammation, leading to metabolic stress, insulin resistance, metabolic syndrome, dislipidemia, cardiovascular disease, and even cancer. Recently, it has been proposed that augmenting energy expenditure (EE), by means of the induction of adaptive thermogenesis in adipose tissue and the increase in the oxidative mitochondrial capacity of muscles, may contribute to alleviating the metabolic stress associated with unhealthy obesity.

The FDA has approved the clinical trials of more and more natural extracts in the past few years, although a basic requirement is to identify the pharmacologically active compounds in combinations that can represent the holistic clinical benefits of the whole extracts.

In a previous work, we have described that PomE (40% punicalagins) augmented the expression of markers of browning and thermogenesis in differentiated human adipocytes, and increased the oxidative respiratory capacity in differentiated human myocytes [8]. Herein, we have conducted a preclinical study of high-fat-diet (HFD)-induced obesity to evaluate the potential of PomE to augment the systemic energy expenditure (EE) as a mechanism to reduce the meta-inflammation and the insulin resistance associated with obesity.

After confirming a prediabetic situation in animals exposed to an HFD for 6 weeks (Appendix A the PomE intervention (HFD-PomE group) or vehicle (HFD group) intervention were initiated in combination with the HFD for twelve weeks. The calculated HOMA/IR index showed a significant reduction in the HFD-PomE group compared to the HFD group (*p*-value 0.047), although without recovering the levels of the ND group (Figure 3), indicating that PomE improves insulin sensitivity after HFD-induced obesity. Interestingly, the HFD-PomE group showed a significant increase in food intake compared to the HFD-control group, although no changes in total body weight nor in the weight of adipose tissues between the two groups were observed. Although we have not quantified the total number of adipocytes, nor their size, in our previous work, PomE augmented lipolysis and markers related to the activation of thermogenesis (*BMP8B*, *UCP1*, *UCP2*) in mature adipocytes, as well as the basal and maximal respiration rates of mature adipocytes compared to nontreated adipocytes. These data suggest a specific role of PomE in the activation of thermogenesis in adipose tissue to augment its function and plasticity.

In addition, by means of indirect calorimetry (Figure 4), an increase in the VO_2_ (*p*-value 0.003), VCO_2_ (*p*-value 0.01), and the EE (*p*-value 0.002) levels in the HFD-PomE group compared to those of the HFD-control group were found. It should be noted that the cages used in this assay do not have any environmental enrichment elements (such as cotton or houses, among others), resulting in a mild cold challenge for the mice. Watanabe’s group [37] has already reported that when there is no environmental enrichment, mice augment the expression of thermogenesis to increase their body temperature [38]. Consequently, it can be concluded that the observed increase in EE without changes in the activity nor in the food intake may be associated with an increase in thermogenesis to compensate the thermal challenge to which the animal is exposed. This effect was further reinforced by the results observed after exposition to cold stress, where the HFD-PomE group displayed a significant increase in rectal body temperature compared to that of the HFD group.

The results of this study also suggest the potential application of PomE extract in other scenarios, such as the improvement of physical exercise, since—as reflected in the literature—there is a great similarity in the thermogenic activation caused by these two conditions [39].

To identify underlying molecular mechanisms implicated in the functional effects of PomE reducing the metabolic stress associated with HFD exposition, the different types of adipose tissue—epidydimal white adipose tissue (eWAT), inguinal adipose tissue (iWAT), and brown adipose tissue (BAT)—and muscle tissues were analysed. Relevant genes implicated in thermogenesis, mitochondrial oxidative capacity, mitochondrial biogenesis, oxidative stress, and β-oxidation were evaluated [9,17,18,19,20,21]. PomE induced the expression of genes related to the early steps of the browning process, such as *BMP8B* and *CPT1A* in BAT; *SIRT1* and *PRDM16* in iWAT; and *BMP8B* and *EPDR1* in eWAT, reflecting the intrinsic differences of the distinct types of adipose tissues (Figure 6).

In addition, after cold exposition, PomE augmented the expression levels of *AMPK*, *SIRT1*, *BMP8B*, and *UCP1* only in the iWAT, which is the one more prone to activating the browning process. AMPK activation is associated to SIRT1 function in the control of key metabolic functions, including differentiation of cultured adipocytes, and it has been extensively associated with the promotion of browning in HFD-induced obesity. Interestingly, the induction of the expression of *SIRT1* by PomE was observed in animals with or without cold exposition, suggesting that PomE extract specifically favours the upregulation of this gene, even in the absence of a low-temperature external stress [11,40]. All in all, these results suggest a relevant role of PomE in the reduction of the atrophy of the AT, in line with the observed reduction in the weight of the eWAT, which has been shown to alleviate systemic inflammation and insulin resistance associated to obesity [22,23,41,42].

Interestingly, PomE also upregulated the expression of the *IL17RA* gene in the iWAT only when exposed to cold. The upregulation of *IL17A* by γδ T cells has been extensively described to activate AT thermogenesis by means of the IL17RA receptor in adipose tissue.

On the other hand, inflammatory myokines released from muscles may contribute to activate lipolysis and browning of AT in a paracrine manner [25], as shown in Figure 7, the expression levels of *IL6* gene in the muscles of mice after cold exposition were statistically increased in the HFD-PomE group compared to the HFD group. Myokines exclusively released by muscles may represent the link from working muscle to other organs such as the adipose tissue, the liver, and the vascular compartments, as suggested by Petersen et al., which also suggests a potential benefit in muscles [26,27].

Although other studies have described pomegranate extracts as improving the metabolic profile in vivo after an HFD by means of the reduction of local and systemic inflammation, IR, or body weight [43,44,45,46,47,48,49], to our knowledge, the effects on energy expenditure or thermogenic program activation were not evaluated.

These results suggest that PomE extract may reduce the negative impact of muscle oxidative stress, increasing the reductive power by means of increased expression of G6PDH, a key target in the formation of the reduced nicotinamide adenine dinucleotide (NADH) [28,29]. NADH’s main function is the exchange of electrons and protons and the production of energy in all cells, which is necessary for proper muscle recovery after physical exercise [30,31], PomE may contribute to maintaining the redox balance, as has been previously reported with other extracts of this fruit [32,33,34,35,36].

In addition, targets related to glucose and redox homeostasis in the skeletal muscle, such as the glucose-6-phosphate dehydrogenase (*G6PDH*) gene, significatively increased after cold exposition in the HFD-PomE group compared to the HFD group. Moreover, a positive tendency was observed in the expression levels of phosphofructokinase (*PFK*) in the BAT, both in normal conditions and after cold exposition, in the HFP-PomE group compared to the HFD group. These results indicate that PomE extract may reduce the negative impact of oxidative stress by increasing the reductive power [28,29,30,31], as reported with other extracts of this fruit [32,33,34,35,36,46,50,51], but also by means of the promotion of WAT browning and BAT activation to increase the systemic energy expenditure.

In summary, PomE used in this study can be postulated as a potential tool against metabolic disorders associated with obesity, by promoting the expression of key markers implicated in the activation of thermogenesis, mitochondrial oxidative capacity, and redox homeostasis. Importantly, PomE diminishes IR, augments the systemic EE, and promotes the maintenance of body temperature under cold stress.

This study opens new possibilities to propose PomE extract as a source of bioactive compounds for the treatment of metabolic chronic diseases. For example, it will be interesting to fractionate the bioactive compounds obtained from pomegranate and to compare distinct combinations of them. In addition, the nutritional status and the genetic susceptibility of individuals will impact on the different responses to the same intervention. Nevertheless, we think this field opens exciting opportunities to alleviate chronic diseases by mean of diet-derived interventions.

## 4. Materials and Methods

### 4.1. Pomegranate Extract (PomE)

PomE (EFSA-approved) was provided by Natac Group company. Bioactive compounds were extracted from the fruit part of *Punica granatum* L.

During the extraction, various solvent ratios of ethanol to water were used (pure water, 50:50, 60:40, and 80:20), with the ratio of ethanol:water 70:30 being the one that released the highest concentration of punicalagins (40%) on one side; and a higher selectivity for punicalagins in contrast to other phytocompound on the other.

### 4.2. Animal Model

To evaluate the potential of PomE in the control of the metabolic stress associated with obesity, a preclinical study of high-fat-diet (HFD)-induced obesity was designed. A total of 28 six-week-old male C57BL/6 mice were divided into three different groups, namely group 1: ND-control (N = 6); group 2: HFD group (N = 10); and group 3: HFD-PomE group (N = 12). High-fat diet (HFD) was obtained from Brogaarden (reference D12451i). The normal diet contained 10% fat, while the high-fat diet contained 45% of total fat. For the first 6 weeks, mice from groups 2 and 3 were fed with the HFD to cause a metabolic disturbance associated with the increased weight. Afterward, at the age of 12 weeks, administration of PomE or the vehicle was initiated. PomE was dissolved in drinking water and administered by gastric gavage 3 days per week during the 12–14 weeks. Drinking water was also administered by gavage as a vehicle to control groups on the same days. Previously, a pilot study was performed to determine the maximal dose of the extract without toxicity. Thus, PomE was administered at a dose of 0.1 g of extract per Kg of mice.

Insulin and glucose tolerance tests, haemograms, and plasma analysis were performed at the beginning, middle, and at the end of the study. Densitometry and indirect calorimetry were performed at the end of the trial, and exposition to the cold chamber for heat stress was performed in the days prior to slaughter.

All procedures were approved by the Research and Animal Welfare Ethics Committee of CNIO-ISCIII Ethics Committee for Research and Animal Welfare (CEIyBA number CBAO6_2018) under the provisions of RD53/2013 law.

Animals were anesthetized with isoflurane 4–5%, followed by a cardiac puncture, to collect the maximum volume of blood.

### 4.3. Glucose Tolerance Test (GTT), Insulin Tolerance Test (ITT), the Index of Homeostasis Model Assessment of Insulin Resistance (HOMA/IR)

#### 4.3.1. Glucose Tolerance Test (GTT)

Previously to the GTT, mice fasted for 16 h. On the day of the test, glucose (0.5 g/mL) was injected intraperitoneally at a dose of 2 g/kg. After injection, blood glucose levels were quantified at the indicated times (0, 15, 30, 60, 120, and 180 min). For this purpose, a superficial cut at the end of the tail was performed with a scalpel, and glucose strips were read by a glucometer (obtained from Menarini Diagnostics, Wokingham, United Kingdom).

Before the injection of glucose, at time 0, about 15–20 μL of blood was extracted and immediately stored at 4 °C for the subsequent analysis of insulin and glucose levels [52].

#### 4.3.2. Insulin Tolerance Test (ITT)

The human insulin was diluted in saline buffer at a concentration of 0.125 U/mL, and the amount of insulin to be injected was adjusted to 0.75 U/kg. After the insulin injection, blood glucose was quantified at different times (0, 15, 30, 60, 60, 120, and 180 min) using the same glucose strips as described before [52].

#### 4.3.3. The Index of Homeostasis Model Assessment of Insulin Resistance (HOMA/IR)

Fasting blood was collected using capillary tubes in the tail´s cut at time 0 min, prior to the glucose injection. Plasmas were obtained after 10 min of centrifugation at 12,000 rcf in a centrifuge at 4 °C. Fasting insulin was determined using the Elisa kit obtained from Crystal Chem (Inc-90080; Downers Grove, IL, USA). Fasting basal insulin and glucose values were used to calculate HOMA/IR, applying the validated equation HOMA/IR = (Fasting Glucose × Fasting Insulin)/22.5 as described [53,54].

### 4.4. Indirect Calorimetry

Biological parameters of mice—the total volume of oxygen (VO_2_) exchanged, the volume of carbon dioxide (VCO_2_) expelled, the energy expenditure (EE), the activity of the mice (movement) and the intake of food and drinking water—from HFD control group and HFD-PomE group were monitored individually, using the metabolic cages of the Oxylet Panlab Harvard Apparatus at the Spanish National Cancer Research Center (CNIO).

Previously, the fluxes of oxygen and carbon dioxide in the metabolic cages were stabilized using the weight of solid tissues determined by dual X-ray absorption densitometry with the GE Medical Systems PIXImus Lunar Densitometer. Images were registered for 5 min for each mouse, taking measurements of the mass of solid and fat tissues. Before the exposition, all animals were anesthetized with 2% isoflurane by inhalation.

After the acclimatization and stabilization of fluxes for 72 h, the study variables were monitored for another 72 h, in cycles of 12 h of light and 12 h of darkness. The analysis was performed separately for repeated measurements in the third day, during the day or the night, given the different physiological behaviour expected in these two periods.

### 4.5. Cold Chambers

A cold chamber with insulated cages was used for individualized mice. During the stay in the chamber, the temperature of each mouse was measured using a rectal probe for three times (on days zero and three and the day of the sacrifice). Mice were individually introduced into the chamber cages at 18 °C for acclimatation for 3 days, thus reducing the heat shock of the mice. After these 3 days, the temperature was gradually reduced to 4 °C. The chamber was maintained at 4 °C for another 24 h. After this time, all mice were sacrificed, and their tissues were collected for the subsequent molecular analysis.

The gavage with vehicle or treatment was performed just before introducing the animals into the chamber, and after the opening of the chamber to measure their temperature before setting it at 4 °C.

### 4.6. Gene Expression Analysis

For the RNA extraction from tissues, the corresponding tissue fragments, frozen at −80 °C, were cut on dry ice using cold forceps and a tempered scalpel (keeping low RNA degradation by the action of RNA-ases). Then, the tissues were homogenized in 1–1.5 mL of phenol over dry ice (TriReagent obtained from Sigma; Burlington, MA, USA), using a T10 IKA homogenizer. Once the tissues were disaggregated, 0.2 volumes of chloroform were added, and the mixes were centrifuged for 15 min at 12,000 rcf at 4 °C. After that, the clear supernatants, corresponding to RNA, were collected and transferred to RNA-ase-free tubes. A total of 0.5 volumes of isopropanol were added and then centrifuged for 10 min (12,000 rcf at 4 °C). Total RNAs were quantified with a UV-Vis spectrophotometer (NanoDrop 2000/2000c, obtained from ThermoFisher Scientific; Waltham, MA, USA).

A total of 1 μg of total RNA of each sample was converted into cDNA with the commercial high-capacity RNA-to-cDNA Master Mix kit from Life Technologies-ThermoFisher; Waltham, MA, USA). The melting curve, on Applied Biosciences 2720 Thermal Cycler equipment, consisted of 10 min at 25 °C, 120 min at 37 °C, 5 min at 85 °C, and finally at 4 °C until storage at −20 °C.

qRT PCRs were performed on the 7900HT Real-Time PCR apparatus using Taqman Gene Expression Master Mix and Taqman-specific probes (Appendix A). Tissue samples were analysed in triplicates. The analysis of gene expression was performed using the Expression Suite Software version 1.0.4 (Thermo Fisher Scientific, Waltham, MA, USA), and the calculation of the relative expression of each gene was performed following the 2^−^^ΔΔ^^Ct^ method [55]. Beta-2-Microglobulin (B2M) or 18S RNA were used as housekeeping genes.

### 4.7. Statistical Analysis

One-way analysis of variance (ANOVA) followed by Bonferroni as a *post-test* was used to determine statistical differences for the analysis of glucose, insulin, and gene expression between groups. All statistically significant values are indicated with asterisks based on their *p*-value. * *p* < 0.05, ** *p* < 0.01, *** *p* < 0.001. All graphical representations show the results as the mean ± standard deviation of the averages of the experiments. The statistical program R version 3.6.1 (Warwick, UK) was used for the comparative analysis of the indirect calorimetry assay. Thus, both the average values and the area under the curve (AUC), obtained by the trapezoidal integration, on the third day were modelled. Linear mixed models were used to appropriately consider the intra-mouse correlation using, on the predictor variable side as a random factor, a categorical variable; and as fixed factors, the weight of solid tissues (previously obtained by densitometry), time 0 h as the baseline, as well as an interaction between a three-level factor corresponding to the day, and a two-level factor corresponding to the treatment (extract versus control). In addition, marginal tests were performed for the significance of the treatment. Excel Professional Plus 2016 software and GraphPad Prism 8.0.1 (La Jolla, CA, USA) were used for the graphical representations.

## Figures and Tables

**Figure 1 ijms-23-10460-f001:**
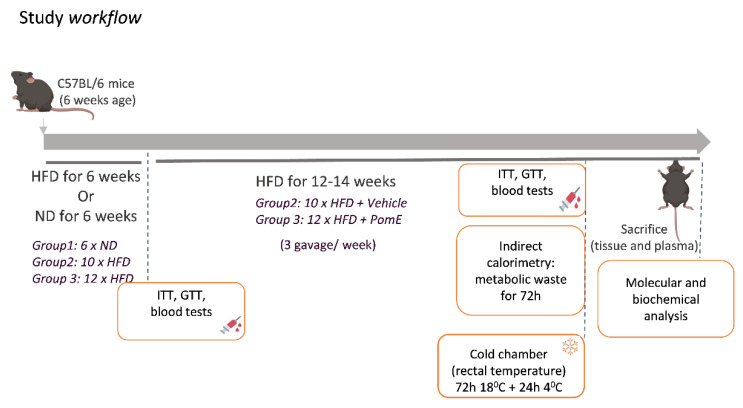
Shows the study workflow indicating the biochemical and molecular determinations (insulin and glucose tolerance tests; densitometry, indirect calorimetry, and the cold chamber exposition for heat stress).

**Figure 2 ijms-23-10460-f002:**
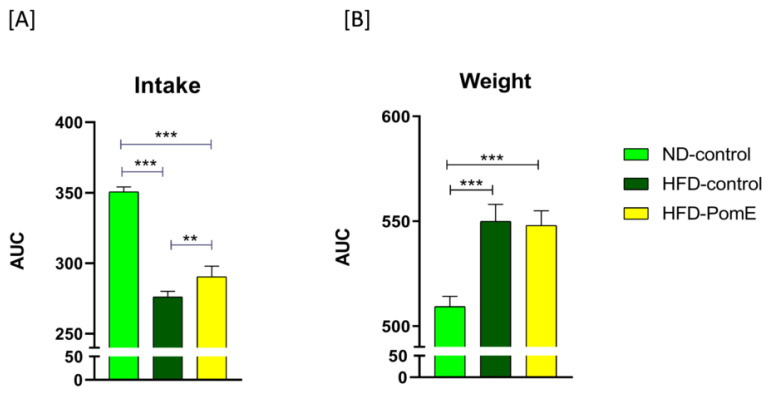
Comparison of food intake (**A**) and weight (**B**) of the animals of ND-control group with that of HFD-control and HFD-PomE groups. Statistically significant values are indicated with asterisks based on their *p*-value, ** *p* < 0.01, *** *p* < 0.001.

**Figure 3 ijms-23-10460-f003:**
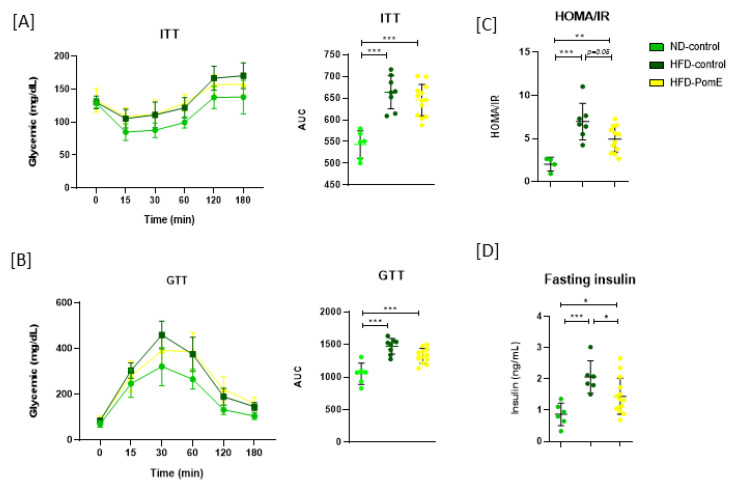
Metabolic phenotyping of glucose and insulin homeostasis in the different groups. Insulin (**A**) and glucose (**B**) tolerance tests were performed at the indicated hours. The area under the curve (AUC) is also represented. (**C**) HOMA/IR values for the different groups. (**D**) Fasting insulin levels for the different groups. Asterisks refer to the comparison with the corresponding control ND group, and between the HFD-PomE and HFD-control groups. Statistically significant values are indicated with asterisks based on their *p*-value, *, *p* < 0.05, ** *p* < 0.01, *** *p* < 0.001.

**Figure 4 ijms-23-10460-f004:**
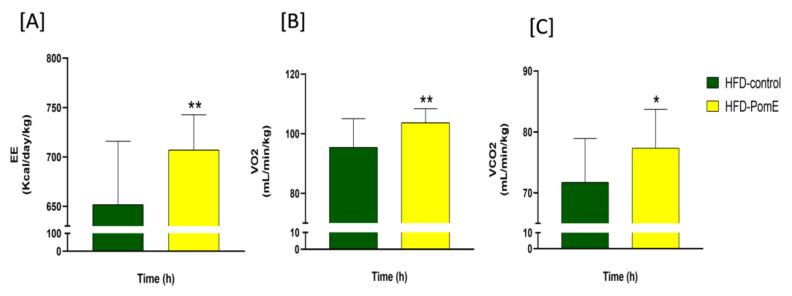
(**A**) Energy expenditure, (**B**) volume of oxygen (VO_2_), and (**C**) volume of CO_2_ (VCO_2_) parameters in HFD-PomE and HFD-control groups (N = 8). Statistically significant values are indicated with asterisks based on their *p*-value, * *p* < 0.05, ** *p* < 0.01.

**Figure 5 ijms-23-10460-f005:**
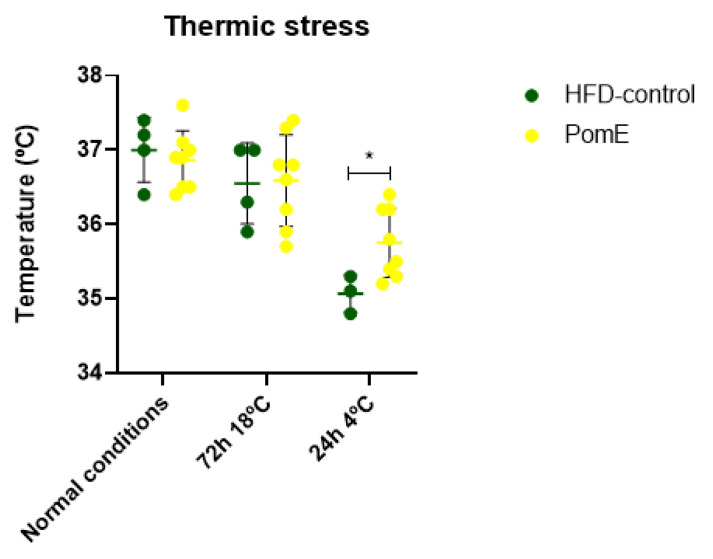
Mice rectal temperature after cold exposition in HFD-PomE (N = 4–6) and HFD groups (N = 4–6). Statistically significant values are indicated with asterisks based on their *p*-value, * *p* < 0.05.

**Figure 6 ijms-23-10460-f006:**
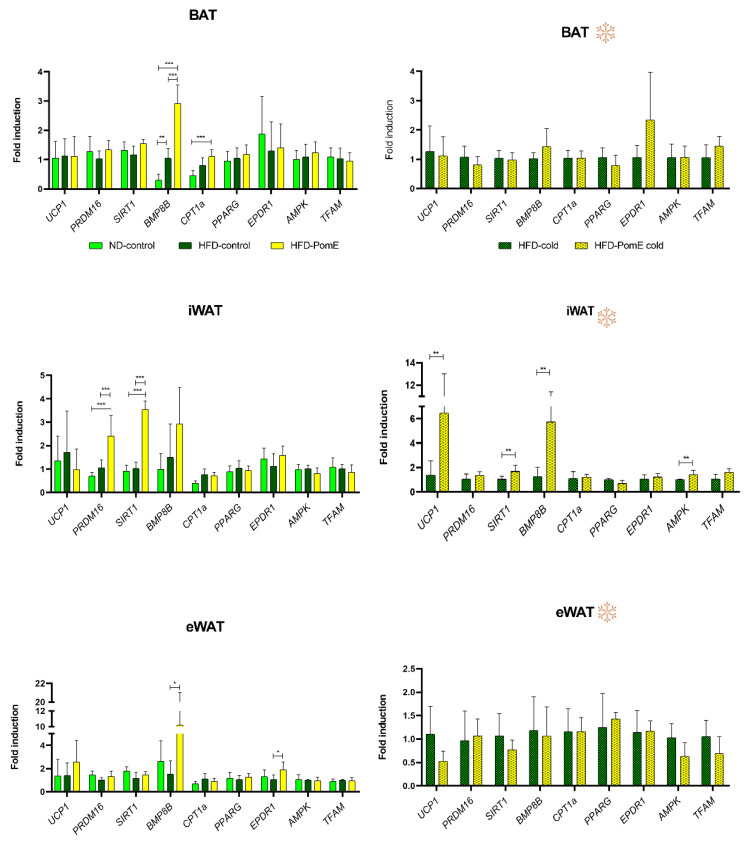
Gene expression levels of targets related to thermogenesis in the different types of adipose tissues (brown (BAT), inguinal white (iWAT), and epididymal white (eWAT) adipose tissues), in ND, HFD-control and HFD-PomE groups, exposed or not to cold stress (N = 4–8). Statistically significant values are indicated with asterisks based on their *p*-value, * *p* < 0.05, ** *p* < 0.01, *** *p* < 0.001.

**Figure 7 ijms-23-10460-f007:**
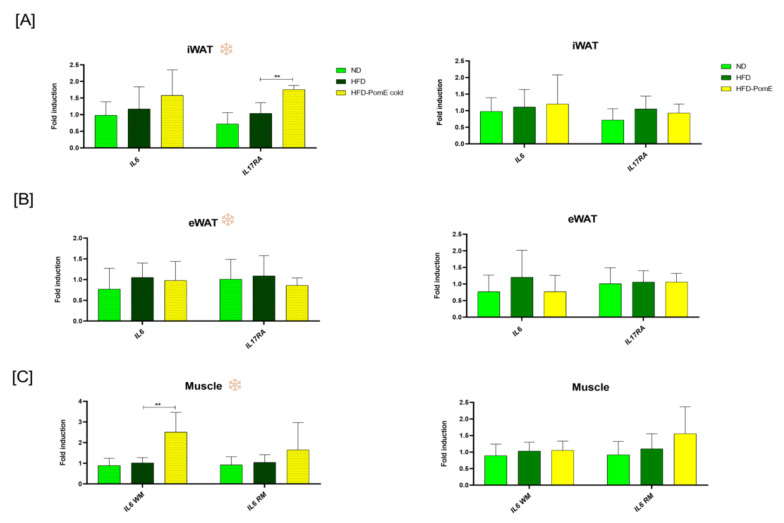
Gene expression levels of the thermogenic *IL17AR* in the (**A**) inguinal white (iWAT) and (**B**) epididymal white (eWAT) adipose tissues, and (**C**) the myokine *IL6* in white (WM) and red (RM) muscle tissues, of HFD-control and HFD-PomE groups, exposed or not to cold stress (N = 4–8). Statistically significant values are indicated with asterisks based on their *p*-value, ** *p* < 0.01.

**Figure 8 ijms-23-10460-f008:**
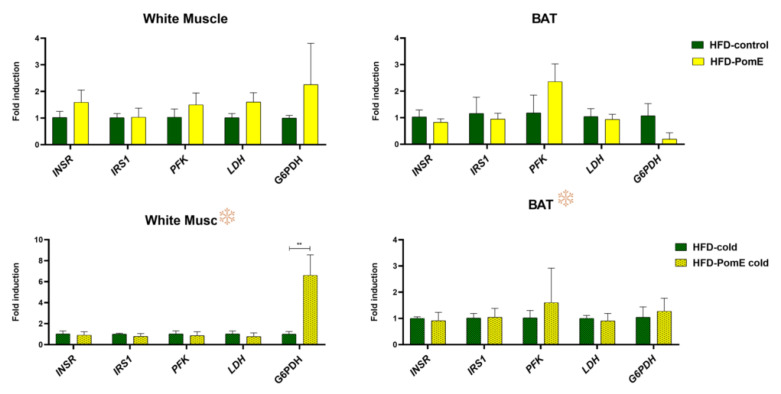
Gene expression levels of genes related to glycolysis and redox homeostasis in brown adipose tissue (BAT) and in the white muscle tissues of HFD-control and HFD-PomE groups, exposed or not to cold stress (N = 4–8). Statistically significant values are indicated with asterisks based on their *p*-value, ** *p* < 0.01.

**Figure 9 ijms-23-10460-f009:**
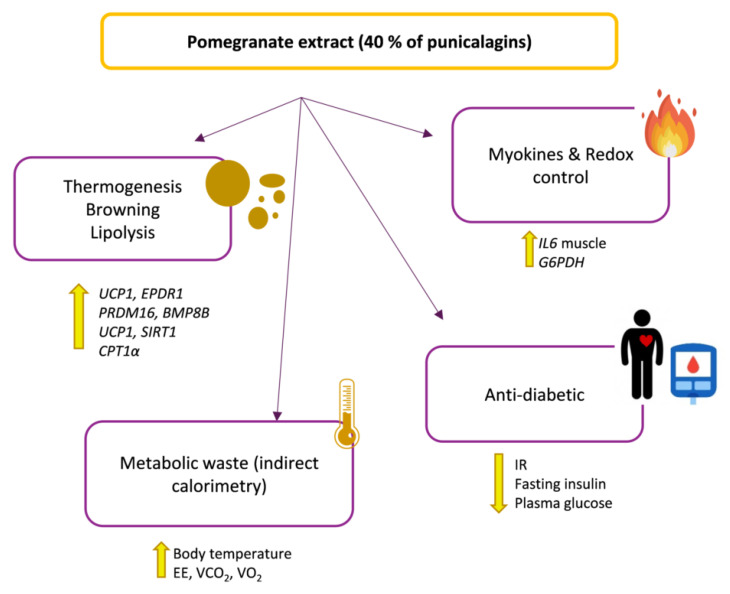
Proposed model of PomE to alleviate the metabolic stress associated to HFD-induced obesity.

## Data Availability

No datasets were generated or analyzed during the current study.

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
