# Peer review of "Pomegranate Extract Augments Energy Expenditure Counteracting the Metabolic Stress Associated with High-Fat-Diet-Induced Obesity"

_ijms, 2022, doi:10.3390/ijms231810460_

Round 1

Reviewer 1 Report

Manuscript # 1863519

In the current article titled “ Pomegranate extract augments energy expenditure counteracting the metabolic stress associated with high fat diet-induced obesity,” the authors have provided the beneficial effect of pomegranate extract to improve metabolic stress and energy expenditure in high fat diet-induced obesity model which is an interesting piece of work to determine the beneficial effect of natural fruit extracts on human physiology and regulate metabolic alteration associated with metabolic diseases including obesity. Although this study has provided insight into Pomegranate extract against the obesity-associated metabolic alteration, however, I have the following comments in this article

Comments:

1)    There are several fruits that have beneficial effects. What was the rationale to use PomE? Why did the authors have chosen PomE? It is fruit extracts and we do not know which component has beneficial effects. If Punicalagin has all these effects then, can we test the effect of Punicalagin separately?

2)    There are other extracts including beetroot extracts, black plum and berries have protective effects on cardiovascular and diabetes. Why have the authors not chosen other fruit extracts including these?

3)    In my opinion, PomE has beneficial effects on obesity, but we cannot rule out the beneficial effect of other plant extracts otherwise PomE would be the next-generation medicine for obese individuals.  Still, we need to characterize the specific component of PomE for its beneficial effect. 

4)    In figure 1, the authors have found that PomE improved HFD food intake, still, it is not causing weight gain in the HFD-PomE group. Do the authors have any explanation?

5)    Did the authors find any changes in the reduction of adipocyte number and lipogenesis?

6)    In line- 232-233 statement, In addition, the weight of epididymal white adipose 232

            tissues (eWAT) of animals in the HFD-PomE group did not increase compared to that of  

            the HFD control group (Supplementary Figure 1), The statement is not matching with     

            Supplementary figure

7)    In figure 3, PomE did not improve GTT and ITT. Do the authors have any explanation? I do not think  PomE improves obesity-associated metabolic alteration.

8)    Authors should represent their bar graph presentation with an individual sample.

9)    In figure 6, browning-associated genes are only activated in normal conditions but not in cold conditions. Ideally, in cold conditions body adapt to regulate body temperature thus expressing the temperature (energy) regulating gene. Any explanation?

10) In figure 6, BMP8B seems significantly high (iWAT) but it is not shown. Any explanation?

11) In your previous paper (In reference 8), AMPK was significantly high after PomE supplementation, however, in this study there is no effect. Did the authors find any change in mitochondrial biogenesis?  

12)  The authors have suggested increased IL17RA  (in iWAT) after PomE supplementation and controlling body temperature. IL-6 increased in WM in cold. How are these cytokines associated with body temperature regulation? Did the authors find any change in lipolysis for energy production during cold along with activation of browning genes?

13)  The authors need to rewrite the manuscript. Results and discussion are mixed. This manuscript has many grammatical/typo errors. Authors should make changes in the title as I have highlighted.

14)  The authors could have done a small-scale human study to see the beneficial effect of PomE. 

Author Response

Answers to Reviewer 1

Authors thank the reviewer for all the comments and suggestions to improve the manuscript.

Manuscript # 1863519

In the current article titled “ Pomegranate extract augments energy expenditure counteracting the metabolic stress associated with high fat diet-induced obesity,” the authors have provided the beneficial effect of pomegranate extract to improve metabolic stress and energy expenditure in high fat diet-induced obesity model which is an interesting piece of work to determine the beneficial effect of natural fruit extracts on human physiology and regulate metabolic alteration associated with metabolic diseases including obesity. Although this study has provided insight into Pomegranate extract against the obesity-associated metabolic alteration, however, I have the following comments in this article

Comments:

1)    There are several fruits that have beneficial effects. What was the rationale to use PomE? Why did the authors have chosen PomE? It is fruit extracts and we do not know which component has beneficial effects. If Punicalagin has all these effects then, can we test the effect of Punicalagin separately?

Thank you for the comment. We agree this should be better clarified.

PomE (40% punicalagins) belongs to the class of quantified extracts (as being defined by the European Medicines Agency (EMA), and the European Pharmacopoeia), where the active marker, i.e. punicalagins, is used to obtain a product with constant quality and reproducible efficacy, although we also know that it is not the only one responsible of the activity. It is not possible to expect an extract to quantify 100% of the ingredients, as it is technically unfeasible.

In this work, we have selected PomE (40% punicalagins) based on our previous screening with 20 natural extracts, EFSA approved for human use, with the objective to identify those with the highest effects on the activation of thermogenesis and mitochondrial oxidative capacity. The extracts were classified on main families depending on the nature of their main bioactive compounds. From this screening, PomE exerted highest effects on these processes both at the molecular level and cell function. Herein, we aimed to validate in vivo the beneficial effects of this extract in the activation of thermogenesis and browning in a high fat diet induced obesity.

With the understanding that the pathogenesis of many diseases involves multiple factors, the focus of drug discovery has shifted from the conventional “one target, one drug” model to a new “multi-target, multidrug” model [1-3]. Although the main bioactive compounds present in PomE are punicalagins, it should be note that additional bioactive compounds may contribute synergistically to the observed effects. It has been shown that natural extracts, as a combination of bioactive compounds, target simultaneously different pathways, which may improve the beneficial effects of the extract. Indeed, accumulating evidence from clinical studies support that natural extracts represent an efficient form of therapy in the control of complex diseases, such as metabolic syndrome, cardiovascular disease, type 2 diabetes and cancer and diabetes [4].

 This has been included in the Discussion section.

2)    There are other extracts including beetroot extracts, black plum and berries have protective effects on cardiovascular and diabetes. Why have the authors not chosen other fruit extracts including these?

Thank you for the comment.

As indicated previously, the previous screening of 20 natural extracts indicated that PomE (40% punicalagins) exerted the highest effects on the activation of thermogenesis and oxidative phosphorylation, both at the molecular level and cell function, in vitro. Herein, we wanted to demonstrate in vivo the beneficial effects of the extract as a complementary approach to alleviate the metabolic stress after a HFD induced obesity by mean of the activation of thermogenesis and browning of the adipose tissue.

This study is part of a predoctoral grant supporting the research in the frame of the industrial program of the Community of Madrid (IND2017/BIO-7826), which promotes the collaboration between Natac Biotech and IMDEA Food institute with the objective to augment our knowledge on molecular targets and mechanism of action of natural extracts in the treatment of chronic diseases. This way, the interest of this program is to apply approved natural extracts to the improve the human health.

3)    In my opinion, PomE has beneficial effects on obesity, but we cannot rule out the beneficial effect of other plant extracts otherwise PomE would be the next-generation medicine for obese individuals.  Still, we need to characterize the specific component of PomE for its beneficial effect. 

We totally agree with this comment. In the frame of precision nutrition, the use of nutritional interventions as complementary approaches in the treatment of metabolic diseases, requires the identification of the main bioactive compounds and their molecular targets and mechanism of action.

 This study opens new possibilities to propose PomE as a source of bioactive compounds for the treatment of metabolic chronic diseases. For example, it will be interesting to fractionate the bioactive compounds obtained from Pomegranate and to compare distinct combinations of them. In addition, the nutritional status and the genetic susceptibility of individuals, will impact on the different responses to the same intervention. Nevertheless, we think this field opens exciting opportunities to alleviate chronic diseases by mean of diet derived interventions. 

This has been included in the final part of the manuscript.

4)    In figure 1, the authors have found that PomE improved HFD food intake, still, it is not causing weight gain in the HFD-PomE group. Do the authors have any explanation? 5)    Did the authors find any changes in the reduction of adipocyte number and lipogenesis?

 Thank you for the comment. This was one of the most surprising results.
As indicated, PomE augmented the HFD intake compared to that of the HFD-Control group. Surprisingly, there were no changes on total body weight, nor in the weight of adipose tissues between the two groups. Although we have not quantified the total number of adipocytes, nor their size, in our previous work, PomE augmented lipolysis and markers related to the activation of thermogenesis (BMP8B, UCP1, UCP2) in mature adipocytes, as well as the basal and maximal respiration rates of mature adipocytes compared to non-treated adipocytes. These data suggest a specific role of PomE in the activation of thermogenesis in adipose tissue to augment its function and plasticity.

This has been included in the manuscript (Discussion).

6)    In line- 232-233 statement, In addition, the weight of epididymal white adipose tissues (eWAT) of animals in the HFD-PomE group did not increase compared to that of the HFD control group (Supplementary Figure 1), The statement is not matching with Supplementary figure

We are truly sorry for the mistake. This has been corrected in the main text of the manuscript (Supplementary Figure 2 instead of Supplementary Figure 1).

“In addition, the weight of epididymal white adipose tissues (eWAT) of animals in the HFD-PomE group did not increase compared to that of the HFD control group (Supplementary Figure 2)”

7)    In figure 3, PomE did not improve GTT and ITT. Do the authors have any explanation? I do not think PomE improves obesity-associated metabolic alteration.

Thank for the comment. As observed in Figure 3, although the GTT and IIT curves were not statistically different between the HFD-PomE and HFD-Control groups, the fasting insulin levels (Figure 3D) were diminished in the HFD-PomE group compared to that of the HFD-Control group, resulting in an improved statistically significant HOMA-IR index in HFD-PomE group (Figure 3D), which suggests a benefit in the control of glucose homeostasis.  

8)    Authors should represent their bar graph presentation with an individual sample.

Thank you for the comment.

We are not sure if you are asking for the ITT, GTT and HOMAIR figure with individual values. Please consider the graph with the individual sample values. As indicated before, although the GTT and IIT curves were not statistically different between the HFD-PomE and HFD-Control groups, the fasting insulin levels (Figure 3D) were diminished in the HFD-PomE group compared to that of the HFD-Control group, resulting in an improved statistically significant HOMA-IR index in HFD-PomE group (Figure 3D), which suggests a benefit in the control of glucose homeostasis. 

For clarification and to maintain the style with the rest of figures, we have maintained this representation.

Reviewer 2 Report

The paper deals with the preclinical study of high fat diet (HFD) induced obesity where PomE augments the systemic energy expenditure (EE) contributing to reduce the low grade of chronic inflammation and insulin resistance associated to obesity. At the molecular level, PomE promotes browning and thermogenesis in adipose tissue, reducing inflammatory markers and augmenting the reductive potential to control the oxidative stress associated to the HFD input.

The purpose of study is described clearly. All methods are completely suitable for this study. All methods are completely suitable for this study. The paper shows valuable results, and the results were clear. I think that the manuscript can be acceptable after minor reversion.

 Materials and Methods Section

 1. Page 2-line 82. In the preparation of Pomegranate extract, please describe the procedure of extraction including the extraction time, extraction temperature and filtration conditions(type of filter, pore size of filter etc.) as a process diagram.

 2. Page 2-line 82. What is the reason that you did extract the Pomegranate under the solvent ratio of ethanol to water=70 : 30? Have you ever tried any other conditions such as 60 : 40, 50 : 50 and 80 : 20? Because the composition and yield of natural plant extract greatly may be affected by the solvent ratio.

Reviewer

Author Response

Authors thank the reviewer for the comments and suggestions to improve the manuscript. Please, consider the answers and the revised version R1 of the manuscript.

The paper deals with the preclinical study of high fat diet (HFD) induced obesity where PomE augments the systemic energy expenditure (EE) contributing to reduce the low grade of chronic inflammation and insulin resistance associated to obesity. At the molecular level, PomE promotes browning and thermogenesis in adipose tissue, reducing inflammatory markers and augmenting the reductive potential to control the oxidative stress associated to the HFD input.

The purpose of study is described clearly. All methods are completely suitable for this study. All methods are completely suitable for this study. The paper shows valuable results, and the results were clear. I think that the manuscript can be acceptable after minor reversion.

 Materials and Methods Section

  1. Page 2-line 82. In the preparation of Pomegranate extract, please describe the procedure of extraction including the extraction time, extraction temperature and filtration conditions(type of filter, pore size of filter etc.) as a process diagram.

Thank you for the comment. We agree this information is relevant as it will impact on the main bioactive compounds obtained from the fruit.

Extraction temperature: 60ºC. Extraction time: 2 hours

Filtration conditions: microfiltration - filter pore size 0.22 micrometres

This has been included in the Materials and Methods section.

  1. Page 2-line 82. What is the reason that you did extract the Pomegranate under the solvent ratio of ethanol to water=70 : 30? Have you ever tried any other conditions such as 60 : 40, 50 : 50 and 80 : 20? Because the composition and yield of natural plant extract greatly may be affected by the solvent ratio.

Thank you for the comment.

The selection of the extraction solvent ethanol to water at a ratio of 70 to 30 was due as it allowed to obtain the highest extraction yield of punicalagins, the active marker of pomegranate extracts. There were tried various solvent ratios of ethanol to water during this phase (pure water, 50:50, 60:40, 70:30 and 80:20) and the solvent that resulted in the highest yield of punicalagins on one side, and a higher selectivity for punicalagins in contrast to other phytocompounds on the other, was the ratio ethanol:water 70:30.

This has been included in the Materials and Methods section.

Round 2

Reviewer 1 Report

The authors have responded to the reviewer's comments adequately. However, for comment 8, the reviewer wanted the author to represent all the bar graphs with individual values (N= x) if possible. This will show the extent of variability in data.  

Author Response

Authors thank the reviewer for the constructive comments to improve the Manuscript.

We have modified figures indicating individual mice values when possible. This means that some of the determinations are not for the individual mice: (1) the food intake was monitored per cage but not the individual mice food intake, and the representation indicates the AUC from all the monitored values; (2) same for the metabolic cages: although the biological parameters -VO2 exchanged, VCO2 expelled, the EE, the activity of the mice (movement), the intake of food and drinking water- and the measurements of the mass of solid and fat (GE Medical Systems PIXImus Lunar Densitometer) were monitored individually every 5 min, the analysis integrates all the data registered for mice included in the different groups as provided by the software Oxylet Panlab Harvard Apparatus.

Please, tell us if this is ok for the reviewer´s comments or if it needs to be better clarified.